# Antibiotic Use and Long-Term Outcome in Patients with Tick-Borne Encephalitis and Co-Infection with *Borrelia Burgdorferi* Sensu Lato in Central Europe. A Retrospective Cohort Study

**DOI:** 10.3390/jcm8101740

**Published:** 2019-10-20

**Authors:** Maša Velušček, Rok Blagus, Tjaša Cerar Kišek, Eva Ružić-Sabljić, Tatjana Avšič-Županc, Fajko F Bajrović, Daša Stupica

**Affiliations:** 1Department of Infectious Diseases, University Medical Center Ljubljana, Japljeva 2, Ljubljana 1525, Slovenia; 2Institute for Biostatistics and Medical Informatics, Faculty of Medicine Ljubljana, Vrazov trg 2, Ljubljana 1104, Slovenia; rok.blagus@mf.uni-lj.si; 3Institute for Microbiology and Immunology Ljubljana, Faculty of Medicine Ljubljana, Zaloška 4, Ljubljana 1000, Slovenia; tjasa.cerar@mf.uni-lj.si (T.C.K.); eva.ruzic-sabljic@mf.uni-lj.si (E.R.-S.); tatjana.avsic@mf.uni-lj.si (T.A.-Ž.); 4Department of Neurology, University Medical Center Ljubljana, Zaloška 2, Ljubljana 1000, Slovenia; fajko.bajrovic@mf.uni-lj.si; 5Faculty of Medicine Ljubljana, Vrazov trg 2, Ljubljana 1000, Slovenia

**Keywords:** tick-borne encephalitis, Lyme borreliosis, antibiotic therapy, co-infection

## Abstract

In this retrospective cohort study of patients with tick-borne encephalitis (TBE), the clinical outcome in relation to co-infection with *B. burgdorferi* sensu lato (s.l.) and, specifically, the effect of antibiotic treatment on clinical outcome in patients with TBE who were seropositive for borreliae but who did not fulfil clinical or microbiologic criteria for proven co-infection, were assessed at a single university medical center in Slovenia, a country where TBE and Lyme borreliosis are endemic with high incidence. Among 684 patients enrolled during a seven-year period from 2007 through 2013, 382 (55.8%) had TBE alone, 62 (9.1%) had proven co-infection with borreliae and 240 (35.1%) had possible co-infection. The severity of acute illness was similar in all the groups. The odds for incomplete recovery decreased during a 12-month follow-up but were higher in women, older patients, and in those with more severe acute illness. Incomplete recovery was not associated with either proven (odds ratio (OR) 1.21, 95% confidence interval (CI) 0.49–2.95; *p* = 0.670) or possible co-infection (OR 0.95, 95% CI 0.55–1.65; *p* = 0.853). Among patients with possible co-infection, older patients were more likely to be prescribed antibiotics, but the odds for incomplete recovery were similar in those who received antibiotics and those who did not (OR 0.82, 95% CI 0.36–1.87; *p* = 0.630), suggesting that routine antibiotic treatment in patients with TBE and possible co-infection may not be warranted.

## 1. Introduction

Tick-borne encephalitis (TBE) and Lyme borreliosis (LB), caused by TBE virus and *Borrelia burgdorferi* sensu lato (s.l.), respectively, are both transmitted by certain species of the tick *Ixodes* and are the most prevalent tick-borne diseases in Europe [1,2]. In countries such as Poland, Russia, and Slovenia [3], with areas where these diseases are endemic, patients with TBE are reported to have borrelial co-infection with frequencies ranging from 13.5% to 16.7% for proven co-infection, 1.9% to 18.8% for borrelial central nervous system (CNS) co-infection, and 7.4% to 46.9% for possible co-infection [4,5,6,7,8,9,10,11,12] (Table 1).

Diagnosis of TBE is straightforward and is based on clinical presentation, cerebrospinal fluid (CSF) pleocytosis, and specific serologic tests [1,13,14], but antiviral therapy is not yet available [1]. In contrast, highly effective antibiotic treatment is available for LB [15] but the interpretation of serologic tests for borrelial antibodies is more complex [2]. The interpretation of *Borrelia* serology is particularly difficult in patients with established diagnosis of TBE because, apart from erythema migrans (EM), which does not necessitate serologic confirmation, the clinical presentations of early LB and TBE may overlap. Thus, differentiating between symptomatic and asymptomatic borrelial co-infection in patients with TBE who test positive for borrelial antibodies represents a diagnostic and therapeutic dilemma, even when laboratory criteria for proven co-infection are fulfilled. Despite these challenges, there has been only limited research on borrelial co-infection in patients with TBE. To our knowledge, data on the long-term outcome in patients with TBE and co-infection with borreliae are limited to one report [5], and there are no data on the effect of anti-borrelial antibiotic therapy in patients with TBE who test positive for borrelial antibodies suggesting possible but not proven co-infection [16].

The aim of this retrospective cohort study was to assess (1) the clinical course and long-term outcome in patients with TBE and proven or possible co-infection with *B. burgdorferi* s.l. and (2) specifically the effect of anti-borrelial antibiotic therapy on the clinical course and long-term outcome in patients with TBE and possible co-infection with *B. burgdorferi* s.l.

## 2. Materials and Methods

The study was approved by the Medical Ethics Committee of the Ministry of Health of the Republic of Slovenia (No. 0120-213/2017-4) and registered at http://clinicaltrials.gov, identifier NCT03958058.

### 2.1. Setting and Patients

Patients ≥18 years old admitted to the University Medical Center Ljubljana, Slovenia, between January 2007 and December 2013 were eligible for the study if they had TBE defined according to European criteria: a febrile illness with symptoms and/or signs of meningitis, meningoencephalitis, or meningoencephalomyelitis, cerebrospinal fluid (CSF) pleocytosis (>5 × 10^6^ cells/L), and demonstration of specific TBE virus IgM and IgG antibodies in serum or intrathecal synthesis of specific antibodies in patients previously vaccinated against TBE [17]. Demographic, clinical, and laboratory data were obtained for evaluation of the severity of acute illness. Patients were assigned to one of three groups according to European defining criteria for LB [16]: (1) TBE infection but no fulfillment of clinical or laboratory criteria for borrelial co-infection (TBE group); (2) TBE infection plus fulfillment of at least one of the following criteria for proven LB: Demonstration of intrathecal synthesis of borrelial IgM/IgG antibodies or isolation of borreliae from CSF (CNS borrelial co-infection), presence of EM on admission or in the preceding or following four weeks, or isolation of borreliae from blood (TBE-LB group); (3) Positive tests for borrelial IgM or IgG antibodies in serum or CSF on admission or up to the 2-month follow-up visit but no fulfillment of criteria for proven co-infection; categorized as TBE and possible borrelial co-infection (TBE-Bb group). Patients were excluded from the analysis if they had a history of past LB or had received antibiotic therapy with anti-borrelial activity for reasons other than borrelial infection on admission or in the preceding four weeks.

### 2.2. Evaluation of Patients

Patients’ medical histories were taken and physical examination performed on admission and on a daily basis during the hospital stay, and at follow-up 2, 6, and 12 months thereafter. In addition, patients were asked, without prompting, an open question about any health-related symptoms that had newly developed or worsened since the onset of TBE. If these symptoms had no other medical explanation they were regarded as TBE-associated symptoms during hospital stay or post-TBE symptoms at follow-up. In clinical examination, particular attention was paid to signs of neurologic involvement (disturbance of consciousness, tremor, ataxia, paralysis, etc.). At discharge from the hospital, the severity of acute illness was assessed quantitatively using a standardized questionnaire as reported previously [18]: presence, intensity, and duration of an individual symptom or sign of TBE were scored on a scale from 0 to 9, and the severity score was defined as the sum of points. In previous evaluation of this system, the scores 0–8, 9–22, and >22 corresponded to clinically mild, moderate, and severe disease, respectively [18].

Complete recovery was defined as the patient being free of new or worsened objective clinical signs attributable to TBE or LB or the patient reporting ≤1 post-TBE symptom. Incomplete recovery was defined as the presence of new or worsened neurologic signs attributable to TBE (post-encephalitic sequelae) and/or the patient reporting ≥2 post-TBE symptoms.

### 2.3. Laboratory Evaluation 

CSF samples collected on admission were analyzed for cell counts and levels of protein and glucose. A leukocyte count > 5 × 10^6^ cells/L (pleocytosis) was considered abnormal. IgM, IgG, and albumin levels were determined in serum and CSF samples. IgM and IgG antibodies to TBE virus were assessed using the Enzygnost^®^ Anti-TBE Virus test (SiemensGmbH, Marburg, Germany). Serologic data for borrelial co-infection were obtained using either a chemiluminescence immunoassay (IgM antibodies to OspC and VlsE, IgG antibodies to VlsE borrelial antigens; Liaison, Diasorin, Italy) or an immunofluorescence assay with a local skin isolate of *B. afzelii* as antigen [18]. Results were interpreted according to the manufacturers’ instructions or as titers, respectively. Liaison borderline results and titers of 1:128 or higher were interpreted as positive [19,20]. Intrathecal synthesis of borrelial antibodies was determined as described by Reiber and Peter for Liaison [21], or by dividing (CSF borrelial IgM or IgG titer/serum borrelial IgM or IgG titer) by (CSF IgM or IgG/serum IgM or IgG). Antibody index values > 1.4 for Liaison or >8 in the immunofluorescence assay were indicative of intrathecal production of borrelial antibodies. Modified Kelly–Pettenkofer (MPK) medium was used for cultivation of *B. burgdorferi* s.l. from blood and CSF samples as described elsewhere [22]. Isolates were identified to species/strain level using MluI restriction of genomic DNA (MluI-length restriction fragment patterns) or by MseI restriction of rrf (5S)–rrl (23S) intergenic spacer amplicons (MseI-restriction fragment-length polymorphism) [23].

### 2.4. Statistical Analysis

Comparisons of the TBE, TBE-LB, and TBE-Bb groups of patients, patients who tested seropositive or seronegative, and patients who did or did not receive antibiotic treatment within the TBE-Bb group, were all analyzed separately, using the Kruskal–Wallis test (comparison of the groups for continuous variables) or Chi-squared test with continuity correction (comparison of groups for categorical variables) as appropriate.

Categorical data were summarized as frequencies (%) and numeric data as medians (interquartile range, IQR). The proportions of patients with incomplete response at 12 months’ post-enrolment across different groups were compared using the normal approximation with continuity correction. The same analysis was used to estimate the proportion of patients with incomplete response at the other time points and at the final evaluable visit.

Association between incomplete response and other covariates, which were determined before data analysis, were estimated using multiple logistic regression. To account for multiple measurements in each patient, random intercept by patient ID was included in the model. Results are presented as odds ratios (OR) with 95% confidence interval (CI); *p* < 0.05 was considered significant. R statistical language; The R Foundation, Vienna, Austria (version 3.4.1) was used for the analyses [24].

## 3. Results

Among the 763 patients with TBE who were hospitalized during the study period, 684 (89.6%) were included in the analysis: 382 (55.8%) had no clinical or microbiologic indicator of borrelial co-infection (TBE group), 62 (9.1%) had proven borrelial co-infection (TBE-LB group), and 240 (35.1%) had possible co-infection (TBE-Bb group).

### 3.1. Patients’ Characteristics on Admission

Baseline data on the severity of acute illness were available for all patients, but not all patients attended follow-up visits (Figure 1). Patients’ basic demographic, clinical and laboratory data at hospital admission are shown in Table 2. The predominant form of TBE was meningoencephalitis (466/684, 68.1%). Serologic test results for borrelial antibodies were available for all patients and were positive in 282/684 (41.2%). The rate of borrelial seropositive results differed according to the serologic test used: serum samples tested using the chemiluminescence immunoassay more often gave a positive result than sera tested in the immunofluorescence assay (174/291, 59.8% and 108/393, 27.5%, respectively). Patients whose sera tested positive for borrelial antibodies were older, more often male, had higher Charlson comorbidity scores, and presented with lower CSF pleocytosis counts than patients with TBE whose sera were negative for borrelial antibodies (Appendix A).

### 3.2. Patients with Tick-Borne Encephalitis (TBE) and Proven Borrelial Co-Infection (TBE-Lyme Borreliosis (LB) Group)

Proven borrelial co-infection was diagnosed in 62 (9.1%) patients: 15 patients had EM, 46 (6.7%) had laboratory diagnosed borrelial CNS infection, two patients had both, and in three patients, borreliae were isolated from the blood. Among 46 patients with borrelial CNS infection, intrathecal synthesis of borrelial IgM or IgG antibodies was demonstrated in 42 patients, five patients had borreliae isolated from CSF, and one patient had both. Among patients with intrathecal synthesis of borrelial antibodies, two patients had concomitant EM. Patients diagnosed with borrelial CNS infection were prescribed intravenous ceftriaxone 2 g once daily for 14 days (36 patients) or oral doxycycline 100 mg twice daily for 14 days (3 patients), and 7/25 patients in whom intrathecal synthesis of borrelial IgM was the only laboratory marker of borrelial co-infection were not prescribed antibiotics. Patients with EM were treated with either intravenous ceftriaxone (8 patients), oral azithromycin (5 patients), or oral doxycycline (2 patients). The three patients in whom borreliae were isolated from blood samples received 14 days of intravenous ceftriaxone. After completing antibiotic therapy, none of the patients with proven borrelial co-infection developed new objective clinical signs of LB during follow-up. In addition, the seven patients with intrathecal synthesis of borrelial IgM who were not prescribed antibiotics did not develop any objective clinical signs of LB during follow-up.

### 3.3. Outcome in Patients with TBE and Possible Borrelial Co-Infection (TBE-Bb Group) According to Anti-Borrelial Antibiotic Therapy

Among the evaluable 684 patients, the 240 (35.1%) patients with possible borrelial co-infection were identified exclusively on the presence of borrelial IgM or IgG antibodies in serum or CSF. Serum tested positive for borrelial IgM, IgG or both IgM and IgG antibodies in 45, 220, and 25 patients, respectively. CSF samples tested positive for borrelial IgM, IgG or both IgM and IgG antibodies in 15, 32, and 6 patients, respectively. Among these 240 patients, 144 (60%) were prescribed anti-borrelial antibiotics and 96 (40%) were not, as decided by attending physicians who were all infectious diseases specialists. Intravenous ceftriaxone 2 g once daily for 14 days was prescribed for 112 (77.8%) patients and oral doxycycline 100 mg twice daily for 14 days was given to 32 (22.2%). Patients who received anti-borrelial antibiotics were older and had higher Charlson comorbidity scores than patients who were not prescribed antibiotics, but the two groups were similar regarding sex, severity of acute illness, and CSF pleocytosis (Appendix A). Patients who were not treated with antibiotics had somewhat higher prevalence of incomplete recovery than patients who were treated, but the difference was not significant (Appendix A). Furthermore, the logistic regression model, adjusting comparison for patients’ age and sex, Charlson comorbidity index, time from enrolment, and severity of TBE, showed that patients who received antibiotics had similar odds for incomplete recovery as those who were not treated (OR 1.23, 95% CI 0.53–2.81; *p* = 0.630, Table 3). The odds for incomplete recovery decreased with time from enrolment and were lower for patients with more numerous comorbidities but higher for those who presented with more severe acute illness (Table 3). During follow-up, none of the patients who qualified as having possible borrelial co-infection developed an objective manifestation of LB that could be attributed to co-infection with borrelial species at the time of TBE virus infection, regardless of whether or not they received antibiotic treatment.

### 3.4. Outcome in Patients with TBE in Relation to Borrelial Co-Infection

At the 2, 6, and 12-month follow-ups, 161/652 (24.7%), 47/462 (10.2%), and 15/294 (5.1%) patients, respectively, had documented new or worsened objective neurologic signs attributable to TBE-post-encephalitic sequelae, such as tremor, limb or bulbar muscles palsies. However, post-encephalitic symptoms, such as headache, fatigue, myalgia/arthralgia, memory or concentration difficulties, were much more frequent than post-encephalitic sequelae, thus incomplete recovery was represented predominantly by the presence of post-encephalitic symptoms at all follow-up visits (Table 4). In the TBE-Bb group, the clinical outcome was similar in patients who were treated with anti-borrelial antibiotics and in patients who were not, therefore, the TBE-Bb group as a whole was compared with the TBE-LB and TBE groups. Regardless of borrelial co-infection, the proportion of patients with incomplete recovery decreased steadily during follow-up, and the differences between the groups were not significant according to univariate analysis (Table 4).

Similarly, the multivariate analysis, which included time from hospitalization, patient age and sex, Charlson comorbidity index, and severity of acute illness, indicated that the odds for incomplete recovery decreased with time from enrolment, and were higher for women and older patients, and lower for patients with more numerous comorbidities, but did not differ between the TBE, TBE-LB, and TBE-Bb groups (Table 5).

### 3.5. Intercurrent Antibiotics, Intercurrent Lyme Borreliosis 

A small proportion of patients received ≥1 course of antibiotics with potential anti-borrelial activity for conditions unrelated to LB during the 12-month follow-up: eight patients (2.1%) in the TBE group, one (1.6%) in the TBE-LB group, and four (1.7%) in the TBE-Bb group. Three patients experienced an intercurrent objective manifestation of LB during follow-up; they were diagnosed and treated for EM by their general practitioners 112, 125, and 189 days after discharge from the hospital, respectively. The first and second patients were from the TBE group and the third was from the TBE-Bb group. All had an uneventful further course of disease.

## 4. Discussion

In this retrospective study of 684 adult patients with TBE, the frequency of proven borrelial co-infection was considerable (9.1%), but possible co-infection was even more frequent (35.1%). We found that neither proven nor possible co-infection with *B. burgdorferi* s.l. was associated with severity of acute illness or long-term clinical outcome in patients with TBE, and specifically that in patients with TBE and possible borrelial co-infection treatment with anti-borrelial antibiotics was not associated with the severity of acute illness or long-term clinical outcome.

The frequency of proven borrelial co-infection in our study (9.1%) was somewhat lower than reported in three earlier studies (13.5% and 16.7%) from the same geographic region and using similar case definitions [4,5,6]. In the present study, proven borrelial co-infection in patients with TBE was defined using rigorous criteria: presence of EM, intrathecal synthesis of borrelial IgM or IgG antibodies, or isolation of borreliae from CSF or blood. If we also took into consideration seroconversion to borrelial antigens as a defining criterion for proven co-infection, as was done in the three earlier studies, then the frequency of proven co-infection in present study would rise to a similar value (13.3%).

Neurologic symptoms and CSF pleocytosis occur in both TBE and Lyme neuroborreliosis; therefore, among patients with TBE and proven borrelial co-infection, only those with EM, which is the certain clinical marker of early symptomatic borrelial infection [15] and does not occur in TBE, can qualify as having definite symptomatic co-infection, i.e. LB. At this point, however, in patients with TBE and laboratory proven co-infection but without EM, it is impossible to differentiate asymptomatic from symptomatic borrelial CNS co-infection, i.e. Lyme neuroborreliosis. 

In our study, EM was found in 2.3% (15/684) of patients with TBE, in line with 1.6% and 2.3% in earlier studies [25]. These frequencies of EM in TBE patients, who by definition must have been bitten by a tick, are within the range of risk estimates of 0.8% to 5.2% for LB after a tick bite in Europe [26,27,28,29], which leads one to assume that potential interactions between TBE virus and *B. burgdorferi* s.l. in relation to co-infection or promotion of co-infection to disease are not clinically significant.

Borrelial CNS co-infection was found in 6.7% and 2.3% of our patients with TBE, depending on the inclusion or exclusion of the 25 patients in whom borrelial CNS co-infection was defined by intrathecal synthesis of borrelial IgM, respectively (since this parameter is reported to have low diagnostic value in Lyme neuroborreliosis [30]), which is in the range of 1.9% to 18.8% in previous reports [4,5,8,9,10,12]. Following the above assumption (that potential interactions between TBE virus and *B. burgdorferi* s.l. with regard to co-infection or promotion of co-infection to disease are not clinically significant), the expected frequency of EM and Lyme neuroborreliosis in patients with TBE should be the same as among LB cases in the general population. Accordingly, since in Europe EM represents >80% of LB cases and <10% Lyme neuroborreliosis [2], among patients in the present study symptomatic borrelial CNS co-infection, i.e. Lyme neuroborreliosis, must have been present at the most in 0.3% (2/684). This estimate would further suggest a very high proportion of asymptomatic borrelial CNS co-infections in patients with TBE and laboratory proven co-infection (95% and 87% depending on the inclusion or exclusion of patients with intrathecal synthesis of borrelial IgM, respectively). This accords with the observation that the large majority of patients with CSF culture-positive *B. afzelii* CNS infection did not fulfil criteria for Lyme neuroborreliosis, indicating that, although *B. afzelii,* which is the predominant causative species of LB in Europe, is able to pass through the blood–brain barrier, it has restricted capability to initiate substantial inflammation of the CNS [31]. On the other hand, intrathecal synthesis of borrelial antibodies may not be detectable shortly after the onset of neurologic symptoms in patients with Lyme neuroborreliosis [2], which leads to false negative laboratory results in a subset of patients. Since, the true proportion of patients with Lyme neuroborreliosis among patients with TBE cannot be determined with certainty, and no data are available on the frequency of asymptomatic borrelial CNS infection in the general population, it is not possible to accurately evaluate any potential interaction between TBE virus and borreliae in the pathogenesis of human CNS co-infection beyond the above estimates. Interestingly, experimental studies in co-infected ticks gave conflicting results, ranging from borrelial species enhancement of TBE virus replication, borrelial species suppression of viral replication, and TBE virus promotion of transmission of borreliae, to no interference between the two pathogens [32].

In our study, the frequency of possible borrelial co-infection (35.1%) was in the range of earlier studies (7.4%–46.9%) [5,6,7,8,9,10,11,12]; however, direct comparisons are not possible because of differences in geographic areas, case definitions, and methods used. The relatively high rate of seropositivity in our study could be explained partly from the follow-up protocol, which enabled us to overcome the low sensitivity of borrelial serologic tests in the first 6–8 weeks after infection [2,15,33], and by defining borderline serologic test results as positive in order to best reflect interpretation of borrelial serology in clinical practice. However, even if patients who showed seroconversion during follow-up (*n* = 29) and those with borderline results (*n* = 46) were removed from the analysis, the frequency of possible co-infection would still be considerable (165/684, 24.1%). It should be pointed out that when interpreting borrelial serologic results, numerous difficulties should be considered [34]; namely, the presence of serum borrelial antibodies can result from current symptomatic but also current asymptomatic infection, or from infection with borreliae in the past, or the test result may be a false positive [34]. In order to lower the possibility that serum borrelial antibodies reflected prior symptomatic infection, we excluded patients who had already experienced LB, but we were unable to identify patients who were seropositive as the result of past asymptomatic infection, or those whose tests gave false positive results. Indeed, in LB endemic areas across Europe, background seropositivity is reported at 8.4%–24% [35], and in Slovenia the rate of IgG positivity in blood donor samples tested using the liaison chemiluminescence assay was 18.4% (9/49) [20]. Accordingly, as in our previous study [36], we found that older age was associated with higher borrelial seropositivity, presumably because the likelihood of encountering borrelial infection in an environment where LB is endemic increases with age. Thus, one can assume that borrelial seropositivity reflects early LB at most in a minority of patients with TBE.

Our results show that in TBE infection, neither proven nor possible borrelial co-infection was associated with the severity of acute illness or unfavorable clinical outcome manifested predominantly as post-TBE symptoms. This is in accord with a few earlier studies [11,12,37], although in one study of 492 patients with TBE, in which 82/489 (16.8%) were diagnosed with CNS borrelial co-infection, the co-infected patients had more severe clinical presentation of acute illness [5]. In another study of 687 patients with TBE, in which 1.9% of patients were diagnosed with CNS borrelial co-infection, the co-infected patients more often presented with neurologic symptoms, more frequently had pleocytosis of >300 cells/mL, and higher CSF protein concentration than those who had TBE alone [9]. However, direct comparisons between these studies are not possible because of differences in geographic areas, case definitions, and methods used.

We found only one study on the long-term outcome in patients with TBE in relation to borrelial co-infection, which showed that patients with TBE and proven borrelial co-infection more often had unfavorable outcomes, but significant differences related only to paresis/paralyses (5/76, 6.0% vs. 6/397, 1.5%) [5]. However, in that study, other covariates were not accounted for in the statistical analysis. In our study, comparing proportions of patients with incomplete recovery at the final evaluable visit and at other follow-up visits implied that patients with complete recovery were less likely to attend follow-up visits, as they are supposedly less motivated to attend check-up appointments. Interestingly, we found that patients with lower Charlson indexes were more likely to have incomplete recovery. Since incomplete recovery was represented predominantly by post-encephalitic symptoms, this finding suggests that patients who were in better health initially were more likely to report such symptoms than patients with more numerous comorbidities. It is possible that comorbidities themselves predisposed patients to already have non-specific symptoms when they contracted TBE infection, which might have raised the threshold for noticing new or increased symptoms since TBE. When persons with chronic diseases are confronted with a new disease, they develop perceptions of the new disease but in addition their former perceptions of chronic comorbidities might change [38].

In our study, patients with TBE and proven borrelial co-infection, but without EM, were almost routinely prescribed antibiotic treatment even though it was not possible to ascertain whether the co-infection represented early LB. Among treating physicians it is usual practice to prescribe antibiotics in order to be on the safe side. Interestingly, the seven patients in whom borrelial co-infection was laboratory proven by demonstration of intrathecal borrelial IgM synthesis, but who were not prescribed antibiotics, did not develop objective manifestations of LB during follow-up. However, the number of patients who were not prescribed anti-borrelial antibiotics in this group was too small to permit statistical analysis of the association between antibiotic use and clinical outcome.

Among patients with possible co-infection, those who received antibiotics had somewhat lower probability of incomplete recovery than those who were not prescribed antibiotics, but the difference was small and not significant. Older patients were more likely to be treated with antibiotics, presumably because older persons with more numerous comorbidities may appear more fragile, lowering physicians’ threshold for prescribing antibiotics. Because, apart from age, we did not find any other characteristics that might have prompted prescribing antibiotics, it is highly unlikely that all the patients in this group who actually had LB were treated with antibiotics. Since none of the patients with possible co-infection, defined regardless of the serologic method used and who were not treated with antibiotics, developed any objective LB manifestation during follow-up, it is also unlikely that the patients who actually had LB were equally distributed between the antibiotic-treated and no antibiotic groups. Therefore, it seems more likely that the patients with TBE who tested positive for borrelial antibodies but did not fulfill criteria for proven co-infection most probably did not have LB. This suggests that the wait-and-see strategy with clinical follow-up, rather than routine antibiotic therapy, may be warranted and safe in these patients. 

Several limitations should be considered when interpreting our findings. First, owing to the retrospective nature of our study, we were unable to account for all potential variables that were included in the decision to prescribe anti-borrelial antibiotic therapy in patients with possible borrelial co-infection. For example, the decision to prescribe antibiotics could be arbitrary across clinicians and influenced by their general comfort level with not prescribing antibiotics to patients with positive tests for borrelial antibodies. Nevertheless, some of the confounding must have been overcome by the similar severity of acute illness in patients who received antibiotics and those who did not. Second, we cannot exclude the possibility that if the more sensitive chemiluminescence assay has been used for all patients, the beneficial effect of antibiotics may have become significant because the larger sample size would have increased the power of the statistical test. However, even if statistically significant, the potentially beneficial effect of antibiotics is expected to be small and, therefore, of low clinical relevance. Third, the post-encephalitic symptoms and their potential impact were not assessed using any validated measure of functioning or quality of life, and objective measures would be preferable. Fourth, our results pertain to a particular geographic area and timeframe and may not be applicable to other locations or other periods. Nevertheless, we believe that management of TBE infections by infectious diseases specialists at a single center with decades of clinical experience and with strict systematic clinical and borrelial serologic follow-up is a particular strength of our study, as was the opportunity to analyze data from a large number of patients that permitted comparison of clinical characteristics between patients who had TBE alone and those who met the strict defining criteria for borrelial co-infection.

## 5. Conclusions

In our retrospective cohort study of patients with TBE in a region such as Slovenia, where TBE is endemic and with relatively high frequency of borrelial co-infection, neither proven nor possible co-infection was associated with the severity of acute illness or the long-term outcome. Most importantly, in patients with TBE who tested positive for borrelial antibodies but who lacked clinical or microbiologic markers of proven co-infection, long-term outcome was not affected by treatment with anti-borrelial antibiotics. This suggests, that vast majority of these patients most probably did not have LB and, therefore, the wait and see strategy may be warranted for them. Whether antibiotic therapy may have a favorable effect in selected patients with TBE and possible borrelial co-infection needs further more controlled studies.

## Figures and Tables

**Figure 1 jcm-08-01740-f001:**
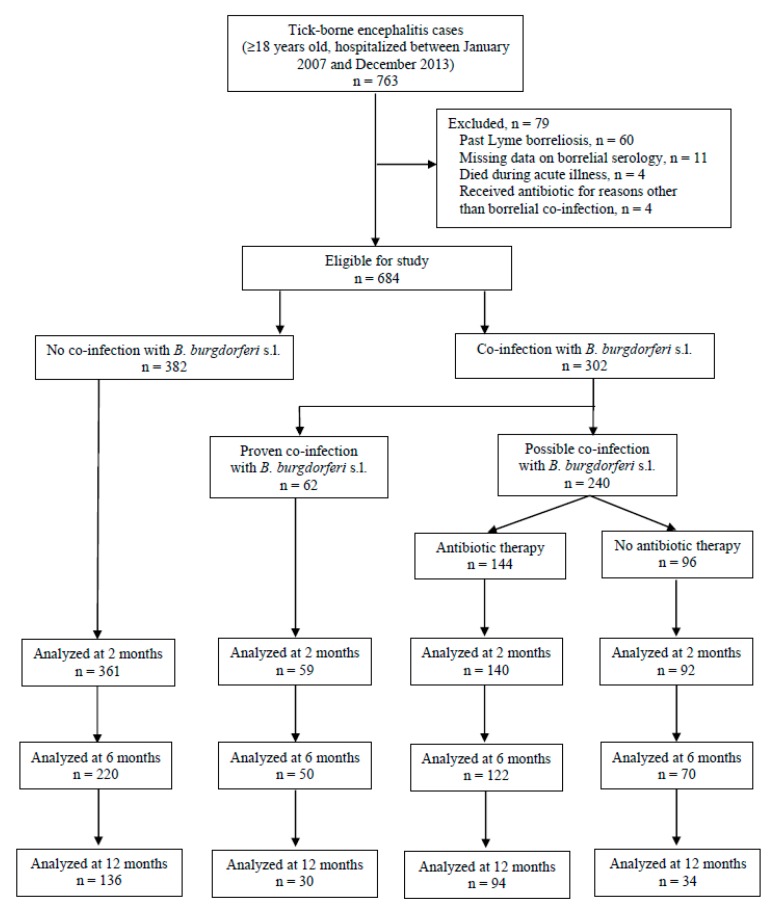
Study diagram.

**Table 1 jcm-08-01740-t001:** Reports on patients with tick-borne encephalitis and co-infection with *Borrelia burgdorferi* sensu lato ^a^.

Study Period Country	Number of Patients and Patients’ Characteristics	Proven Borrelial Co-Infection	Proven Borrelial CNS Co-Infection ^b^	Possible Borrelial Co-Infection ^c^	Proven and Possible Borrelial Co-Infection
1992–1993 Slovenia [4]	89 patients with TBE	12 (13.5)	6 (6.7)	/	/
1994 Slovenia [5]	492 patients with TBE	82 (16.7)	61/489 (12.5)	60/484 (12.4) ^d^	/
1995–1996 Slovenia [6]	36 patients with TBE out of 130 patients with acute febrile illness after a tick bite	6 (16.7)	/	3 (8.3) ^d^	9 (25)
1999–2001 Russia [7]	445 patients with TBE out of 1952 patients with acute febrile illness after a tick bite	/	/	33 (7.4) ^e^	/
1995–2004 Slovenia [8]	32 patients with TBE	/	6 (18.8) ^f^	15 (46.9) ^g^	/
1993–2008 Poland [9]	687 patients with TBE	/	13 (1.9)	116 (16.9) ^h^	/
2003–2009 Slovenia [10]	11 patients with peripheral facial palsy out of 1218 patients with TBE	/	1 (9.1)	2 (18.2) ^d^	3 (27.3)
2009–2012 Poland [11]	110 patients with TBE	/	/	30 (27) ^i^	/
2007–2012 Slovenia [12]	717 patients with TBE	/	22/661 (3.3)	66/655 (10.1) ^g^	/

Data are *n* (%) or *n*/*n* (%). Abbreviations: CNS, central nervous system; TBE, tick-borne encephalitis; /, data not available. ^a^ PubMed literature search using the queries “tick-borne encephalitis AND Lyme” and “tick-borne encephalitis AND *Borrelia*” with no limits for year of publication and written in English. Case reports were not included. ^b^ Defining criteria for Lyme neuroborreliosis: isolation of *B. burgdorferi* s.l. from cerebrospinal fluid or intrathecal synthesis of IgG or IgM antibodies specific for *B. burgdorferi* s.l. ^c^ Defined by positive serologic test results. ^d^ Immunofluorescence assay detecting serum IgM and IgG antibodies. Titers of 1:256 were interpreted as positive. ^e^ enzyme-linked immunosorbent assay (ELISA) detecting serum IgM antibodies and immunofluorescence assay with corpuscular antigen Ip-21 strain *B. afzelii* for detection of serum IgG antibodies. ^f^ Chemiluminescence immunoassay (Liaison, Diasorin, Italy) or enzyme immunoassay (IDEIA, DakoCytomation, Denmark). ^g^ Chemiluminescence immunoassay (Liaison, Diasorin, Italy) detecting serum IgG. ^h^ ELISA (Abbot, USA and Biomedica, Austria until 2006, subsequently Virotech, Germany). ^i^ Borrelial co-infection evaluated using whole blood PCR for *Borrelia* species.

**Table 2 jcm-08-01740-t002:** Demographic, clinical, and laboratory characteristics of patients with tick-borne encephalitis on admission according to absence or presence of proven or possible borrelial co-infection.

Characteristic	TBE *n* = 382	TBE-LB *n* = 62	TBE-Bb *n* = 240	*p* Value ^a^
Age	49 (35–61)	53 (43–68)	59 (46–69)	<0.001
Male sex	208 (54.5)	39 (62.9)	148 (61.7)	0.143
Charlson comorbidity index	1 (0–2)	1 (0–2)	2 (0–3)	<0.001
Vaccinated against TBE	14 (3.7)	2 (3.2)	12 (5.0)	0.670
Clinical presentation				0.615
Meningitis	91 (23.8)	16 (25.8)	50 (20.8)	
Meningoencephalitis	259 (67.8)	38 (61.3)	169 (70.4)	
Meningoencephalomyelitis	32 (8.4)	8 (12.9)	21 (8.8)	
Severity of acute illness				0.649
Mild	99 (25.9)	16 (25.8)	54 (22.5)	
Moderate	232 (60.7)	38 (61.3)	145 (60.4)	
Severe	51 (13.4)	8 (12.9)	41 (17.1)	
Severity score of acute illness	12 (8–17)	12 (6.8–17.8)	12.5 (9–19)	0.324
CSF leukocyte count (× 10^6^/L)	104 (54.8–192)	101 (65.3–154)	68.5 (35–134.3)	<0.001

Data are median (interquartile range) or number (%) of patients. Abbreviations: TBE, tick-borne encephalitis without borrelial co-infection; TBE-LB, TBE with proven borrelial co-infection; TBE-Bb, TBE with possible borrelial co-infection; CSF, cerebrospinal fluid. ^a^ Overall *p* value for comparisons between groups was estimated using Kruskal–Wallis test or Chi-squared test with continuity correction as appropriate: *p* < 0.05 was considered significant.

**Table 3 jcm-08-01740-t003:** Factors associated with incomplete recovery in patients with tick-borne encephalitis and possible borrelial co-infection according to antibiotic therapy (yes versus no).

	OR (95% CI) ^a^	*p* Value ^b^
Intercept	0.05 (0.00–0.51)	0.011
Antibiotic therapy (yes vs. no)	1.23 (0.53–2.81)	0.630
Time		
6 vs. 2 months	0.24 (0.13–0.44)	<0.001
12 vs. 6 months	0.50 (0.25–0.98)	0.043
Sex (male vs. female)	0.47 (0.20–1.08)	0.075
Age, years	1.05 (1.00–1.10)	0.058
Charlson comorbidity index	0.53 (0.32–0.87)	0.012
Severity score of acute illness	1.07 (1.02–1.13)	0.007

Abbreviations: OR, odds ratio for incomplete response; CI, confidence interval. ^a^ Estimated from a multiple logistic regression model with incomplete recovery as the dependent variable, adjusted for patient effect. Each OR is adjusted for all other variables in the table. ^b^
*p* < 0.05 was considered significant.

**Table 4 jcm-08-01740-t004:** Number (%) of patients with tick-borne encephalitis who had incomplete recovery at follow-up visits according to absence or presence of proven or possible borrelial co-infection.

	All *n* = 684	TBE *n* = 382	TBE-LB *n* = 62	TBE-Bb *n* = 240	*p* Value ^a^
2 months post-hospitalization	363/652 (55.7)	202/361 (56.0)	32/59 (54.2)	129/232 (55.6)	0.9698
6 months post-hospitalization	170/462 (36.8)	82/220 (37.3)	16/50 (32.0)	72/192 (37.5)	0.757
12 months post-hospitalization	95/294 (32.3)	44/136 (32.4)	10/30 (33.3)	41/128 (32.0)	0.991
At final evaluable visit	219/653 (33.5)	129/361 (35.7)	15/59 (25.4)	75/233 (32.2%)	0.257

Abbreviations: TBE, tick-borne encephalitis; TBE-LB, proven co-infection with *B. burgdorferi* s.l.; TBE-Bb, possible co-infection with *B. burgdorferi* s.l. Overall *p* value for comparisons between groups was estimated using the normal approximation with continuity correction: *p* < 0.05 was considered significant.

**Table 5 jcm-08-01740-t005:** Factors associated with incomplete recovery according to absence or presence of proven or possible borrelial co-infection (proven co-infection vs no co-infection and possible co-infection versus no co-infection).

	OR (95% CI) ^a^	*p* Value ^b^
Intercept	0.05 (0.01–0.22)	<0.001
TBE-LB vs. TBE	0.82 (0.34–2.00)	0.670
TBE-Bb vs. TBE	1.05 (0.61–1.83)	0.853
Time		
6 vs. 2 months	0.21 (0.14–0.31)	<0.001
12 vs. 6 months	0.48 (0.30–0.76)	0.002
Sex (male vs. female)	0.51 (0.31–0.86)	0.011
Age, years	1.04 (1.01–1.07)	0.012
Charlson comorbidity index	0.57 (0.41–0.80)	<0.001
Severity score of acute illness	1.09 (1.06–1.13)	<0.001

Abbreviations: OR, odds ratio for incomplete response; CI, confidence interval; TBE, tick-borne encephalitis; TBE-LB, TBE with proven borrelial co-infection; TBE-Bb, TBE with possible borrelial co-infection; CSF, cerebrospinal fluid. ^a^ Estimated from a multiple logistic regression model with unfavorable outcome as the dependent variable, adjusted for patient effect. Each OR is adjusted for all other variables in the table. ^b^
*p* < 0.05 was considered significant.

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
