# Peer review of "Antibiotic Use and Long-Term Outcome in Patients with Tick-Borne Encephalitis and Co-Infection with Borrelia Burgdorferi Sensu Lato in Central Europe. A Retrospective Cohort Study"

_jcm, 2019, doi:10.3390/jcm8101740_

Round 1

Reviewer 1 Report

The study shows that a coincidence of TBE infection with an infection by Borrelia b.s.l. may occur. Apart from manifest EM, the objective demonstration of later LB manifestations serologically is dubious. This is emphasized in the discussion. The possible coincidental borrelia infection in TBE patients can be treated according to existing guidelines with good clinical outcome. This paper provides a framework for clinicians in desicion-making.

Author Response

Author's reply to the Review Report (Reviewer 1)

Comments and Suggestions for Authors

Reviewer: The study shows that a coincidence of TBE infection with an infection by Borrelia b.s.l. may occur. Apart from manifest EM, the objective demonstration of later LB manifestations serologically is dubious. This is emphasized in the discussion. The possible coincidental borrelia infection in TBE patients can be treated according to existing guidelines with good clinical outcome. This paper provides a framework for clinicians in decision-making.

Answer: We thank the reviewer for his/her comments.

Reviewer 2 Report

General comments:

Recently we observe a growing popularity of not-verified diagnostic methods used in Lyme disease, which is partly caused by limitations of well-established two-step diagnostic approach. Hence, use of antibiotics in patients not fulfilling official criteria for Lyme disease is frequent. The study tackles with an important clinical dilemma, whether such patients with unconfirmed Lyme disease benefit from treatment or not. The important strength of the study is a large cohort (over 600 patients) that was evaluated. However, to appreciate the work better, major improvement and re-writing of the manuscript is necessary. 

I wish the study was prospective.

Unfortunately the group diagnosed with CNS borrelial infection is very diverse. Patients were treated with different antibiotics. Some were not treated at all. In my opinion it is difficult to judge about outcome of the disease (main aim of the study) if there is such a diversity in treatment. This study group should be more homogenous.

The group with suspected Lyme.

The authors did not find any significant differences in patients with suspected borreliosis treated with antibiotics and not treated over a long observation period.

There are only 3 explanations for that:

1) Lyme disease is a disease which does not require antibiotic treatment (obviously not true)

2) All the patients with suspected Lyme in both groups (treated and untreated) were free from the disease or a proportion of sick persons was random and identical (quite likely)

3) The authors could somehow identify sick persons and treat only sick persons with a success (no inclusion criteria were provided “decided by attending physicians who were all infectious diseases specialists” – line 193) (very unlikely)

Could this group be just NO LYME disease?

The authors mention couple times in the manuscript about “asymptomatic borrelial CNS infection”. Finally (line 299) they say “there are no data available on the frequency of asymptomatic borrelial CNS infection in the general population”

To figure out such frequency one should perform lumbar puncture randomly in healthy population which will never ever happen. I do not think anyone can recognize “asymptomatic borrelial CNS infection”

Minor remarks:

Line 64

Change „antimicrobial” to „antiviral”.

Line 65

Try rephrasing the sentence “In contrast, highly effective…”, which is too long and difficult to understand.

Lines 71, 93, 120, 173 and 357

“Microbiologically diagnosed borrelial CNS infection” – did the authors mean “laboratory diagnosed/confirmed”?

Line 95

Please mention which method was used for the direct detection of B. burgdorferi. What does isolation mean?

Table 2

In the table one can find numbers and medians. No statistical test provided. What does p value mean – which values differ one from another?

Line 191

Could the authors provide information how many serum and CSF samples were positive for B. burgorferi in the group with possible infection? Was IgM or IgG more frequently positive?

Paragraph 3.4 is a bit chaotic. Data included there should be explained better. It is not clear how authors define post-encephalitic symptoms and post-encephalitis sequelae. A valuable addition would be if the authors briefly described neurologic signs that were attributed to TBE in follow-ups.

Table 4 and line 232

Time from enrolment sounds as if the patients were enrolled prospectively, whereas this is a retrospective study. “Time from hospitalization” should be used instead.

In the “Discussion” section the authors need to present in a better way the new knowledge generated by the study. Avoid providing results here. All the results should be included in the “Results” section. Consider re-writing this paragraph to better appreciate the work.

EM is not always recalled by patients with LB. The authors judge about interactions of two pathogens based on prevalence of the disease in TBE patients in Slovenia and patients bitten by tick in other parts of Europe (namely The Nederlands – ref. no 24). Unfortunately tick attachment duration or prevalence of Borrelia in ticks may vary in different populations and areas and may affect prevalence of the disease in people. Assumption about lack of interaction between the pathogens (line 276) is far-fetched and should not be made based on that data.

Author Response

Author's reply to the Review Report (Reviewer 2)

Comments and Suggestions for Authors

Reviewer: General comments:

Recently we observe a growing popularity of not-verified diagnostic methods used in Lyme disease, which is partly caused by limitations of well-established two-step diagnostic approach. Hence, use of antibiotics in patients not fulfilling official criteria for Lyme disease is frequent. The study tackles with an important clinical dilemma, whether such patients with unconfirmed Lyme disease benefit from treatment or not. The important strength of the study is a large cohort (over 600 patients) that was evaluated. However, to appreciate the work better, major improvement and re-writing of the manuscript is necessary.

Answer: We appreciate the comments of the reviewer and would like to do our best to address them in order to improve the manuscript.

Reviewer:  I wish the study was prospective.

Answer: We agree with the reviewer, that a prospective study design would make our conclusions more substantiated.

Reviewer: Unfortunately, the group diagnosed with CNS borrelial infection is very diverse. Patients were treated with different antibiotics. Some were not treated at all. In my opinion it is difficult to judge about outcome of the disease (main aim of the study) if there is such a diversity in treatment. This study group should be more homogenous.

Answer: We agree with the reviewer that the group diagnosed with proven borrelial CNS infection is diverse in regard to antibiotic treatment. Of the 46 patients with proven borrelial CNS infection, 36 were treated with intravenous ceftriaxone, 3 with doxycycline and 7 patients, in whom intrathecal synthesis of borrelial IgM was the only microbiologic marker of borrelial co-infection, were not treated with antibiotics. However, both, ceftriaxone and doxycycline are recommended treatment regimens for early Lyme neuroborreliosis (Borg, R.; Dotevall, L.; Hagberg, L.; et al. Intravenous ceftriaxone compared with oral doxycycline for the treatment of Lyme neuroborreliosis. Scand. J. Infect. Dis. 2005; 37: 449–454, Ljøstad, U.; Skogvoll, E.; Eikeland, R.; et al. Oral doxycycline versus intravenous ceftriaxone for European Lyme neuroborreliosis: a multicentre, non-inferiority, double-blind, randomised trial. Lancet. Neurol. 2008; 7: 690–695), and the seven patients who were not treated with antibiotics were the patients in whom CNS co-infection was defined by intrathecal synthesis of borrelial IgM, which is reported to have low diagnostic value in Lyme neuroborreliosis. Therefore, we believe that antibiotic selection should not have had significant influence on treatment outcome in the treated group.

Reviewer: The group with suspected Lyme.

The authors did not find any significant differences in patients with suspected borreliosis treated with antibiotics and not treated over a long observation period.

There are only 3 explanations for that:

1) Lyme disease is a disease which does not require antibiotic treatment (obviously not true)

2) All the patients with suspected Lyme in both groups (treated and untreated) were free from the disease or a proportion of sick persons was random and identical (quite likely)

3) The authors could somehow identify sick persons and treat only sick persons with a success (no inclusion criteria were provided “decided by attending physicians who were all infectious diseases specialists” – line 193) (very unlikely)

Could this group be just NO LYME disease?

Answer: We very much agree with the reviewer that in the group with possible Lyme borreliosis, the treated and untreated patients were either free from the disease or a proportion of sick persons was random and identical in treated and untreated subgroups. This was also one of the conclusions of our study. Actually, it was exactly the reviewer’ question “Could this group be just no Lyme disease” which prompted us to perform this retrospective study because the tool to identify “sick persons” is lacking.

Namely, in clinical practice, the attending physicians are left with the dilemma on how to interpret positive Borrelia serologic results in patients with TBE who lack clinical or laboratory evidence of proven borrelial co-infection but the possibility of concomitant Lyme neuroborreliosis cannot be excluded. Such patients are therefore diagnosed with possible Lyme borreliosis because the clinical presentation of TBE may mask clinical presentation of concomitant Lyme neuroborreliosis. This differs from the clinical scenario of patients who are Borrelia seropositive but without CNS symptoms/signs and pleocytosis.

We amended the manuscript accordingly. Please, see Discussion, lines 391-402, lines 438-440.

Reviewer: The authors mention couple times in the manuscript about “asymptomatic borrelial CNS infection”. Finally (line 299) they say “there are no data available on the frequency of asymptomatic borrelial CNS infection in the general population”

To figure out such frequency one should perform lumbar puncture randomly in healthy population which will never ever happen. I do not think anyone can recognize “asymptomatic borrelial CNS infection”

Answer: We agree with the reviewer that there will be no data available on the frequency of asymptomatic borrelial CNS infection in the general population. Therefore, we amended the manuscript according to reviewer’s comment. Please, see lines 324, 325, 327. However, asymptomatic borrelial CNS infection is known from other populations. In our previous study of patients with multiple erythema migrans, we found that a small proportion of patients had CSF pleocytosis but were asymptomatic in regard to CNS symptoms (Stupica et al. Comparison of Clinical Course and Treatment Outcome for Patients With Early Disseminated or Early Localized Lyme Borreliosis. JAMA Dermatol. 2018;154(9):1050-1056). Such patients could be regarded as having asymptomatic borrelial CNS infection. Still, there is no therapeutic dilemma in these patients because they present with erythema migrans. Similarly, only 20% of patients with B. afzelii isolated from CSF had CSF pleocytosis, not fulfiling criteria for Lyme neuroborreliosis (Strle et al. Comparison of findings for patients with Borrelia garinii and Borrelia afzelii isolated from cerebrospinal fluid. Clin. Infect. Dis. 2006, 43, 704–710). However, these studies were performed on strictly defined patient populations and not in healthy people.

Minor remarks:

Reviewer: Line 64

Change „antimicrobial” to „antiviral”.

Answer: Corrections done. Please, see line 64.

Reviewer: Line 65

Try rephrasing the sentence “In contrast, highly effective…”, which is too long and difficult to understand.

Answer: Correction done. Please, see lines 66 and 67.

Reviewer: Lines 71, 93, 120, 173 and 357

“Microbiologically diagnosed borrelial CNS infection” – did the authors mean “laboratory diagnosed/confirmed”?

Answer: The reviewer correctly pointed out/noticed that by “microbiologically diagnosed” we meant “laboratory diagnosed” as opposed to “clinically diagnosed”. We believe that in this context, “laboratory” is a hypernym in relation to “microbiologically”. We amended the manuscript as suggested by the reviewer. Please, see lines 72, 95, 122, 185, 192, 290, 314, 322, 384.

Reviewer: Line 95

Please mention which method was used for the direct detection of B. burgdorferi. What does isolation mean?

Answer: Corrections done. Please, see lines 136-140.

Reviewer: Table 2

In the table one can find numbers and medians. No statistical test provided. What does p value mean – which values differ one from another?

Answer: Corrections done. Please, see lines 145-147 and 180-182.

Reviewer: Line 191

Could the authors provide information how many serum and CSF samples were positive for B. burgorferi in the group with possible infection? Was IgM or IgG more frequently positive?

Answer: Additional information was added to the manuscript as suggested by the reviewer. Please, see lines 204-206.

Reviewer: Paragraph 3.4 is a bit chaotic. Data included there should be explained better. It is not clear how authors define post-encephalitic symptoms and post-encephalitis sequelae. A valuable addition would be if the authors briefly described neurologic signs that were attributed to TBE in follow-ups.

Answer: The study definition of post-encephalitic symptoms and post-encephalitic sequelae is provided in the Methods section, lines 109-111 and 120-121, respectively. Information was added to the manuscript as suggested by the reviewer. Please, see lines 233-235.

Reviewer: Table 4 and line 232

Time from enrolment sounds as if the patients were enrolled prospectively, whereas this is a retrospective study. “Time from hospitalization” should be used instead.

Answer: Correction done. Please, see Table 4 and line 249.

Reviewer: In the “Discussion” section the authors need to present in a better way the new knowledge generated by the study. Avoid providing results here. All the results should be included in the “Results” section. Consider re-writing this paragraph to better appreciate the work.

Answer: As suggested by the reviewer, parts of the Discussion section were rewritten. Correction done. Please, see lines 271-274, 300-309, 392-412, 438-440.

Reviewer: EM is not always recalled by patients with LB. The authors judge about interactions of two pathogens based on prevalence of the disease in TBE patients in Slovenia and patients bitten by tick in other parts of Europe (namely The Nederlands – ref. no 24). Unfortunately tick attachment duration or prevalence of Borrelia in ticks may vary in different populations and areas and may affect prevalence of the disease in people. Assumption about lack of interaction between the pathogens (line 276) is far-fetched and should not be made based on that data.

Answer: We agree with the reviewer that prevalence of Borrelia in ticks differs across Europe, yet we are aware of only tick attachment duration variations between B. burgdorferi sensu stricto in relation to B. afzelii, but not among different Borrelia species in Europe and B. afzelii is far the most prevalent causative species of erythema migrans in Europe. The risk estimates of 0.8% to 5.2% for Lyme borreliosis after a tick bite refer to different parts of Europe (Sweden, Finland, Switzerland) besides the Netherlands, and represent an average estimate because even within individual countries, pronounced inter-region variabilities may exist, but all of the listed countries report high cumulative incidence rate of Lyme borreliosis. We amended the manuscript accordingly by adding additional references to support this. In addition, we changed our assumption from “there is no interaction between TBE virus and B. burgdorferi s.l. in relation to co-infection or promotion of co-infection to disease” to “which leads one to assume that potential interactions between TBE virus and B. burgdorferi s.l. in relation to co-infection or promotion of co-infection to disease are not clinically significant”. Please, see Ref 27-29, lines 521-529.

Round 2

Reviewer 2 Report

I think the manuscript can be accepted in the current form.

The authors have included a list of limitations of their study so the reader is aware of them.

I personally think that sentences (394-400) are too long. The sentence structure is too complicated to understand. Maybe the authors could rephrase them.

“Because, apart from age, we did not find any other characteristics that might have prompted prescribing antibiotics, it is highly unlikely that all the patients in this group who actually had LB were treated with antibiotics. Since none of the patients with possible co-infection, defined regardless of the serologic method used and who were not treated with antibiotics, developed any objective LB manifestation during follow-up, it is also unlikely that the patients who actually had LB were equally distributed between the antibiotic-treated and no antibiotic groups.”